

# Changes in sprint performance and sagittal plane kinematics after heavy resisted sprint training in professional soccer players

Johan Lahti[1], Toni Huuhka[2], Valentin Romero[3], Ian Bezodis[4], Jean-Benoit Morin[1,5,6] and Keijo Häkkinen[2]

[1] LAMHESS, Université de Nice-Sophia Antipolis, Nice, France
[2] Neuromuscular Research Center, Biology of Physical Activity, Faculty of Sport and Health Sciences, University of Jyväskylä, Jyväskylä, Finland
[3] Centre for Sport Studies, Universidad Rey Juan Carlos, Madrid, Spain
[4] Cardiff School of Sport and Health Sciences, Cardiff Metropolian University, Cardiff, United Kingdom
[5] UJM-Saint-Etienne Interuniversity Laboratory of Human Movement Biology, Université de Lyon, Saint-Étienne, France
[6] Sports Performance Research Institute New Zealand (SPRINZ), Auckland University of Technology, Auckland, New Zealand

Corresponding author
Johan Lahti,
johan.lahti1@etu.unice.fr,
lahti.johan87@gmail.com

## ABSTRACT

**Background**. Sprint performance is an essential skill to target within soccer, which can be likely achieved with a variety of methods, including different on-field training options. One such method could be heavy resisted sprint training. However, the effects of such overload on sprint performance and the related kinetic changes are unknown in a professional setting. Another unknown factor is whether violating kinematic specificity via heavy resistance will lead to changes in unloaded sprinting kinematics. We investigated whether heavy resisted sled training (HS) affects sprint performance, kinetics, sagittal plane kinematics, and spatiotemporal parameters in professional male soccer players.

**Methods**. After familiarization, a nine-week training protocol and a two-week taper was completed with sprint performance and force-velocity (FV) profiles compared before and after. Out of the two recruited homogenous soccer teams ($N = 32$, age: $24.1 \pm 5.1$ years: height: $180 \pm 10$ cm; body-mass: $76.7 \pm 7.7$ kg, 30-m split-time: $4.63 \pm 0.13$ s), one was used as a control group continuing training as normal with no systematic acceleration training (CON, $N = 13$), while the intervention team was matched into two HS subgroups based on their sprint performance. Subgroup one trained with a resistance that induced a 60% velocity decrement from maximal velocity ($N = 10$, HS60%) and subgroup two used a 50% velocity decrement resistance ($N = 9$, HS50%) based on individual load-velocity profiles.

**Results**. Both heavy resistance subgroups improved significantly all 10–30-m split times ($p < 0.05$, $d = -1.25$; $-0.62$). Post-hoc analysis showed that HS50% improved significantly more compared to CON in 0–10-m split-time ($d = 1.03$) and peak power ($d = 1.16$). Initial maximal theoretical horizontal force capacity (F0) and sprint FV-sprint profile properties showed a significant moderate relationship with F0 adaptation potential ($p < 0.05$). No significant differences in sprinting kinematics

or spatiotemporal variables were observed that remained under the between-session minimal detectable change.

**Conclusion**. With appropriate coaching, heavy resisted sprint training could be one pragmatic option to assist improvements in sprint performance without adverse changes in sprinting kinematics in professional soccer players. Assessing each player's initial individual sprint FV-profile may assist in predicting adaptation potential. More studies are needed that compare heavy resisted sprinting in randomized conditions.

## INTRODUCTION

Sprinting performance has been shown to be effective in distinguishing different levels of soccer players (*Haugen et al., 2014*; *Cometti et al., 2001*). Accordingly, it makes sense that there exists an interest in finding optimal methods to improve sprint performance in high level settings (*Haugen et al., 2014*). This likely also explains the fact that articles on soccer and sprinting have increased exponentially in the last two decades (*Nikolaidis et al., 2016*). However, there still seems to be a lack of sprint performance intervention articles, especially in professional settings. Therefore, researching the usefulness of different training options for sprint performance enhancement within a professional soccer setting seems warranted.

One option that may provide a beneficial stimulus for sprint performance is resisted sprint training (*Kawamori et al., 2014*; *Bachero-Mena & González-Badillo, 2014*; *Morin et al., 2017*; *Pareja-Blanco, Asián-Clemente & SáezdeVillarreal, 2019*; *Cross et al., 2018*; *Alcaraz et al., 2018*; *Alcaraz, Elvira & Palao, 2014*; *Spinks et al., 2007*; *Cahill et al., 2019*). Different forms of resisted sprint training have been used with the aim to improve sprint performance by overloading different parts of the sprint acceleration phase, both from a intermuscular coordination and structural standpoint (*Cahill et al., 2019*). Recently, there has been a growing interest in exploring the value of heavy resistance in assisting improvements in sprint performance (*Morin et al., 2017*; *Pareja-Blanco, Asián-Clemente & SáezdeVillarreal, 2019*; *Cross et al., 2018*). Based on the available literature, a definitive definition for heavy resisted sprinting does not seem to exist. One definition for heavy resistance could be that it prioritizes within moderation overloading kinetic properties (force application) over kinematic specificity (technical similarity). Thus, this would be considered "specific traditional overload" (*Brearley & Bishop, 2019*). According to cross-sectional biomechanical studies, this corresponds to all loads clearly decreasing maximal velocity capacity more than 10% (*Alcaraz et al., 2008*). This has also been reported to be around a less accurate measure of 7.5–15% of body mass (BM), a method that is highly biased towards frictional components and does not consider the relative strength of the athlete (*Cross et al., 2019*). The idea behind heavy loading is to focus on the early acceleration phase of the Force-Velocity (FV) spectrum. Thus from a kinetic standpoint, the focus is on highly overloading the horizontal component of the resultant ground

reaction force vector (*Morin et al., 2017*; *Cotter et al., 2013*; *Kawamori, Nosaka & Newton, 2013*). This stimulus could affect to different degrees both mechanical effectiveness of the ground force orientation during the step (i.e., what ratio of anterior-posterior and vertical forces is the resultant force built upon) and absolute force output, which could lead to improved sprint performance.

Interventions with heavy loads have shown mixed results, possibly to some degree due to different methodology. Four studies showed positive effects on early sprint performance (*Kawamori et al., 2014*; *Bachero-Mena & González-Badillo, 2014*; *Morin et al., 2017*; *Cahill et al., 2020*), another showed split time improvements between 10–30-m, while instead a lighter load group improved also at 0–20-m (*Pareja-Blanco, Asián-Clemente & SáezdeVillarreal, 2019*), and one study showed trivial to small effects on performance from both heavy and light resisted sprinting (*Cross et al., 2018*). Evident methodological differences include large differences in what is considered heavy (range ∼20%–50% velocity decrement), not standardizing each subjects load to a specific velocity decrement (using the less accurate % of BM method) (*Petrakos, Morin & Egan, 2016*), using 1 vs. 2 training sessions per week, initial level and amount of familiarization of subjects, and timing between training completion and post-testing and associated tapering (*Morin et al., 2020*). Limitations have also been discussed, such as not considering each subjects degree of loading needs in terms of initial sprint FV-characteristics in the start of the study (*Cross et al., 2018*).

Furthermore, potential negative effects of violating kinematic specificity by using heavy resistance in sprinting have also been discussed in literature (*Alcaraz et al., 2018*; *Alcaraz, Elvira & Palao, 2014*; *Alcaraz et al., 2019*). These discussions have possibly created uncertainty among coaches, with regards to whether such immediate kinematic and spatiotemporal changes would then lead to detrimental long-term transference to unloaded sprinting. One theory is that training with increased loading may lead to excessive trunk lean (*Alcaraz, Elvira & Palao, 2014*), or create a biomechanically less optimal lower body mechanics, such as excessive flexion (*Alcaraz et al., 2019*). However, only two intervention studies have addressed the long-term effects of resisted sprint training on technique and both using only light resistance (7.5–10% velocity decrement), while comparing to a unresisted sprint training group (*Alcaraz, Elvira & Palao, 2014*; *Spinks et al., 2007*). Despite the light loading, both interventions showed that resisted sprint training led to a very slight increase in trunk lean during initial acceleration, while one of the studies showed that even the unresisted group increased trunk lean (*Spinks et al., 2007*). Increased trunk lean has been associated with improved force production in the anterior-posterior direction (*Atwater, 1982*), thus making it less clear when it is a unwanted adaptation and whether it is dependent on the training modality. Therefore, one possible explanation for why the unresisted group in *Alcaraz, Elvira & Palao (2014)* did not increase trunk lean could be related to the fact that there was no improvement in early acceleration performance, unlike the unresisted group in *Spinks et al. (2007)*. However, adaptations to kinematics should be carefully interpreted to whether it is a cause or an effect and as such may not be directly related.
Therefore, the aim of this study is to investigate changes in sprint performance and the potential underlying mechanical changes (kinematics, spatiotemporal variables, ground force orientation efficiency, and main kinetic outputs) after integrating two different heavy resisted sprint training loading protocols within a professional soccer setting. The aim of the first heavy load is to follow the same maximal mechanical power parameters as in previous literature, which corresponds to a 50% velocity decrement relative to maximal velocity (*Cross et al., 2018*; *Cross et al., 2019*). The aim of the second heavy load is to have a slightly higher focus on maximal strength and early acceleration, which corresponds to a 60% velocity decrement. Our first hypothesis was that both heavy loads will improve early split-time sprint performance, with the heavier load being even more effective at early acceleration. Our second hypothesis was that both loads will increase early acceleration center of mass (CM) distance and CM angle at toe-off.

## MATERIALS AND METHODS

### Study design and participants

A pre-test versus post-test experimental design utilizing three groups was used to examine the effects of heavy resisted sprint training in professional male soccer players. 32 male professional soccer athletes from two teams in the premier division in Finland volunteered to participate in the study using convenience sampling (age: $24.1 \pm 5.1$ years: body-height: $180 \pm 10$ cm, body-mass: $76.7 \pm 7.7$ kg). The sample size in this study was highly similar to previous resisted sprint training studies using comparable methods (*Alcaraz, Elvira & Palao, 2014*; *Spinks et al., 2007*). Inclusion criteria included being a professional soccer athlete competing within the Finnish Premier soccer league. An exclusion criterion was placed for goalkeepers due to the lower amounts of linear sprinting. No exclusion criterion was placed for age, but under 18-year-old athletes were required to have parental consent. Both teams were in initial pre-season and trained on average of 7–10 sessions per week (which included strength training twice per week) and competed an average of once per week. More detailed scheduling can be found in the Tables S9, S10. One professional soccer team was used as two intervention groups and the other professional soccer team as a control group. The soccer team selected to function as the control group did not train early or late acceleration separately from sport-specific practice in their pre-season protocol, including no resisted sled training. Therefore, they were instructed to continue training as normal. The intervention team was further randomly matched into two homogenous subgroups in terms of sprint performance with different heavy sled loading schemes. These loading schemes corresponded either to a heavy sled (HS) load that decreased the athlete's maximal velocity by 50% (HS50%) or 60% (HS60%). A total of 15 training opportunities were provided within 9 weeks (Fig. 1). Including two training sessions each week was not possible because of the teams scheduling conflicts. This corresponded to 6 out of 9 weeks including two sessions per week. Furthermore, tapering was initiated on week 10 and continued to week 11 where post testing was performed. Therefore, both the control and intervention group were tested for sprint performance and kinematic changes 11 weeks apart. Testing was performed on the same day of the week (end of the week, after

Pre-tests and familiarization — Week -1, Week 0     Intervention — Week 1 – 9     Post-tests — Week 11

| | | | Sled sprints | | Sled sprint distance | | Free sprints (20 m) | |
|---|---|---|---|---|---|---|---|---|
| | | Week(s) | Day 1 | Day 2 | HS60 % | HS50 % | Day 1 | Day 2 |
| Familiarization to Sled sprints 80 % of BM x 3 → 20 meters | Load-velocity profiles 1 x 25,50, 75,100 % of BM → 15-25 m | 1 | | 4 | | | 2 | 2 |
| | | 2-4 | 2 | 4 | | | 2 | 2 |
| | | 5-6 | 3 | 5 | 15 m, RECO: 2-3 min | 20 m, RECO: 2-3 min | 1 | 1 |
| Sprint FV-profiles 2x30 m sprints | Sprint FV-profiles 2x30 m sprints | 7 | 5 | Camp # | | | 1 | 1 |
| | | 8 | 2 | | | | 2 | 2 |
| | | 9 | 3 | 5 | | | 1 | 1 |
| | | Taper: 10 – 11 | 2 | | | | | 2 |

Post-tests: Sprint FV-profiles 2x30 m sprints

**Figure 1** **Training program design.** HS: Heavy Sled, *: sled velocity verification was completed on week 1, filming of sled technique on week 2, RECO: recovery time between sprints, m: meters, FV: Force-velocity, #: camp training included two sprints with rubber bands and 2×2 free sprints on separate days.

a low intensity day), but one week apart. The intervention groups had the opportunity to complete two weeks of pretesting on sprint performance and technique analysis, while due to scheduling issues, the control group was available for one week of testing. All training and testing sessions were completed inside on artificial turf, with an exception made for post testing, which was performed outside on the same type of artificial turf on the same time and day of the week. Wind conditions were still $(1 \text{ ms}^{-1})$ on the outdoor post testing day with a highly similar temperature (14 vs. 15 C). Written informed consent was obtained from all athletes on the first day of familiarization, and approval for this study was granted by the University of Jyvaskyla Ethical Committee and was performed in the accordance with the Declaration of Helsinki.

## GROUP ALLOCATION

Athletes in the intervention soccer team were ordered from the lowest to highest 30-m split times derived during two weeks of familiarization and, thereafter, matched in a pairwise manner into either of the following heavy sled groups: HS50% or HS60% to balance variance. The best 30-m performance was used from the two familiarization weeks. The 0–30-m split time was used as it has a lower measurement error compared to smaller split-times (*Haugen, Breitschädel & Samozino, 2018*), and because it was the maximal split-time distance used in our testing protocol. There was no ordering of the control group, however, the sprint performance was predicted to be similar due to earlier research collaboration work with the team involving sprint performance testing. The initial aim was to recruit an equal amount of soccer athletes within the control team. However, only 13 were available to volunteer and were considered healthy by the team physiotherapist to perform sprint testing at this point of the early pre-season. The final group size and respective highly homogenous 30-m performance times were the following:

HS60%, $N = 10$, 4.65 s, 95% CI [4.55–4.77] vs. HS50%, $N = 9$, 4.62 s, 95% CI [4.56–4.69] vs. CON, N: 13, 4.63 s, 95% CI [4.55–4.70], $p = 0.88$.

## Testing procedures and data analysis
### Sprint Force-Velocity profile and performance tests
Following warm-up, all participants completed two 30-m maximal sprints from a standing stance start. The passive recovery between sprints was three minutes. Sprint performance (split times 0–5, 0–10, 0–20, and 0–30-m), kinetic outputs and mechanical efficiency were computed pre- and post-training from the best time trial. Data was derived from a radar device (Stalker ATS Pro II, Applied Concepts, TX, USA), using a validated field method as reported previously (*Haugen, Breitschädel & Samozino, 2018*; *Samozino et al., 2016*; *Morin et al., 2019*). Individual linear sprint Force-Velocity (FV) profiles in the antero-posterior direction were calculated and thereafter relative theoretical maximal force (*F0:* N.kg$^{-1}$), velocity (*v0*: m s$^{-1}$), and maximal power (Pmax: *W*.kg$^{-1}$) capabilities. Despite the use of an approximate measurement of "maximal power", which can be considered a pseudo-power (*Vigotsky et al., 2019*), the term maximal power output will be used in this study. Mechanical efficiency was calculated based on the maximal ratio of forces (RFmax in %) and the average ratio of forces for the first 10-m (Mean RF on 10-m in %). These RF values are a ratio of the step-averaged horizontal component of the ground-reaction force to the corresponding resultant force, i.e., these values aid the interpretation of mechanical effectiveness with which the ground force is oriented in early acceleration (*Morin & Samozino, 2016*). RFmax depicts the theoretical maximal effectiveness of directing force forwards in the first step of the sprint (within the constraints of sprint running stance, the higher the value of RFmax, the more forward, horizontally-oriented the ground push during the stance phase). Mean RF on 10-m focuses on the same parameter, but is an average of the forward force application effectiveness over the first 10-m. A more horizontally oriented ground reaction force was considered beneficial within the range of values reported in this study.

### Load-velocity tests
The final sled familiarization session was combined with load-velocity testing. Load-velocity tests were completed using one unresisted and three resisted sprints (50%, 75%, 100% of BM) for both HS groups, outlined in previous literature (*Cross et al., 2017*). Thereafter, individualized load-velocity profiles were created for each athlete with a least-square linear regression (*Cross et al., 2017*). The individual resistance leading to a 60% and 50%-velocity decrement from maximal velocity was calculated.

Sled velocity was verified with the radar on the first week of training to be within a 5% range of the targeted velocity. A total of 3 athletes' loads had to be modified with an increase of 2.5–7.5 kg, that were verified again the following week (Final ranges, HS60%: −58.4%, 95% CI [−59.4−−57.5], HS50%: −49.4%, 95% CI [−51.4−−47.5]).

## Sprint spatiotemporal and kinematics assessment
For all FV-profile sprints, video images were obtained at 240 Hz with a smart phone video camera at a HD resolution of 720p (Iphone6, Apple Inc, Cupertino, Ca). The kinematic sprint sequences of interest were the touchdown (first frame the foot was visibly in contact

with the ground) and toe-off (first frame the foot had visibly left the ground) across the first extension and three steps of early acceleration and 3 steps in upright sprinting of the sprint using 6× zoom in Kinovea (v.0.8.15), similar to previous literature (*Wild et al., 2018*). The same leg sequence was analyzed pre-post, with a secondary effort to analyze the sequence as close to the midpoint of the camera as possible. The cameras were placed 9-m perpendicular at the 1.5-m mark and the 22.5-m mark along a 0– 30-m line, at a 1.1 m height, allowing approximately a 9-m field of view. 1.5-m was chosen based on that the first three steps have been considered unique to early acceleration (*Von Lieres und Wilkau et al., 2018*), taking place within around three meters in this population. Upright mechanics were analyzed at 22.5-m based on that team sport athletes are at around 95% or at maximal velocity at this phase (*Clark et al., 2019*).

Furthermore, an additional data analysis was performed in the second week of the study to observe the immediate effects of the resisted sprint training on early acceleration mechanics. The second week was chosen so that the athletes had time to react to the used coaching cues, which are defined in the intervention section. According to our data, sleds at this resistance magnitude reach maximal velocity around 5-m, therefore going into a velocity maintenance phase for the remaining meters (∼10-m for HS60%, ∼15-m for HS50%). Thus, this was considered the main stimuli zone for each sprint, and therefore, it was used to compare to early acceleration of the unloaded sprint. This was done by having the sled sprint start 5-m before the calibration zone for unloaded early acceleration.

All filming zones were calibrated to a 5-m horizontal distance along the midpoint of the camera at the line. The human body was modelled as 18 points. This required manual digitization of the following: vertex of the head, halfway between the suprasternal notch and the 7th cervical vertebra, shoulder, elbow, and wrist joint centers, head of third metacarpal, hip, knee, and ankle joint centers, and the tip of the toe.

The following spatiotemporal and kinematic step characteristics were determined after exporting the digitalized coordinates to Excel (Microsoft Office 2016): contact time (s), step length (m; horizontal displacement between initial contact of one foot and the point of initial contact of the opposite foot, measured from the toe tips), and step rate (Hz; calculated as 1/step time, where step time was determined as the sum of contact time and the subsequent aerial time). Whole-body center of mass (CM) location was calculated using *De Leva (1996)* segmental data. This allowed for the calculation of touchdown and toe-off distances (horizontal distance between the toe and the CM, with positive values representing the toe ahead of the CM). Furthermore, angles of the trunk (relative to the horizontal) and the hips (ipsilateral and contralateral) were quantified. All distances of CM were normalized to the height of the athlete and reported as (m/body length) (*Wild et al., 2018*). All sprints were analyzed twice to improve reliability with the digital marker method.

### Intervention

Training protocols are outlined in Fig. 1. Familiarization within the intervention group for sled training was initiated two weeks before the training intervention and was combined with the sprint Force-Velocity (FV)-profile tests (2 × 30-m sprints), including group

allocation based on sprint performance. A load of 80% of BM ($2 \times 15$-m sprints) was selected for familiarization. A total of 15 heavy resisted sprint training session opportunities were planned within 9 weeks and an additional two-week taper (two sessions total) across the 11-week pre-season. This 11-week interval included a break week in the form of an international training camp. Therefore, resisted sprint training sessions were, in general, twice per week, transitioning from a total of six resisted sprints per week up to eight at the midway point (week 5). All training sessions included 20-m free sprints, which were in the start of the program two per session, transitioning to one free sprint per session after the midway point. All athletes were harnessed at their waist, using the 21 kg sprint sleds (DINOX, customized sled, Finland). To standardize the stimuli between athletes within both intervention subgroups, a velocity-based training approach was utilized, where all athletes used a load that adapted their velocity to the desired threshold. In this case HS60% used a load leading to a 60% velocity decrement from maximal velocity and HS50% used a load leading to a 50% decrement from maximal velocity. The 50% load was chosen to simulate power properties as it has been shown that external maximal power is reached approximately at 50% of maximal velocity in a maximal acceleration sprint (*Cross et al., 2017*). The heavier 60% velocity decrement load was chosen with the aim to stay within proximity to the 50% load but stimulate more maximal strength properties, thus an even higher bias towards early acceleration. On the artificial training surface, this 10% velocity difference corresponded to the average relative mass of 120% of BM in the HS60% group and 94% of BM in the HS50% group (including the mass of the sled), equating to a group average difference of 26 kg. A sled sprint distance of 0–15-m for the HS60% group and 0–20-m for the HS50% group was used to standardize sprint time (HS60%: 4.26 s, 95% CI [3.74–4.77], HS50%: 4.73 s, 95% CI [4.39–5.08], $p = 0.15$). Training was supervised by the team strength and conditioning coach and completed after the warm-up for technical and/or tactical training on field. Pre-training warm-up ($\sim$15 min) included light running, dynamic full-body stretches, muscle and dynamic movement pattern activation, and low to high intensity sprint exercises. Between-sprint rest was three minutes. Both groups were given the same coaching cues, that is, prioritizing stride power (or push) over stride frequency and high arm movement with aligned posture. Finally, post testing was completed at the end of a two-week tapering period, by reducing the modality specific volume down from eight sprints a week to two, with one session of two free sprints per week.

## Statistical analysis

Shapiro–Wilk's test was used to test the data's normality and levene's test was used to examine the homogeneity of variance. A $3 \times 2$ (group $\times$ time) repeated-measured ANOVA with Bonferroni post-hoc comparisons was used to determine the within- and between-group effects as well as examining interaction effects. Baseline measures were used as covariates to control for the effect of initial sprint performance. Sprint performance was defined mechanically (Pmax, $F_0$, RFmax, Mean RF on 10-m, $v_0$, and Sprint FV-profile), by split-times (5-m, 10-m, 20-m, and 30-m), spatiotemporally (contact time, step rate, step length at initial acceleration and maximal velocity), and kinematically (hip angle, trunk angle, CM distance). For each individual the sprint with the best 30-m time within pre

and post testing was compared statistically for both mechanical-, split times- and sprint technique variables. Independent and paired two-tailed t-tests were used to analyse within- and between-group differences of the immediate effects of the resisted sprint training on early acceleration mechanics (two groups). Given the large number of analyses (26), we adjusted for multiple comparisons using the Benjamini–Hochberg procedure utilizing a false discovery rate of 0.05 (*Benjamini & Hochberg, 1995*). Effect sizes (ES) were calculated using pooled SD and interpreted with Hopkins' benchmarks to distinguish small ($\geq$0.2), moderate ($\geq$0.6), large ($\geq$1.2) effects (*Hopkins, 2002*). Accounting for typical fluctuations in athletes' weekly sprint performance and sprint technique was of interest in our study. Thus, minimum detectable change (MDC) with 95% confidence intervals was calculated from the difference in best performance sprint FV-profile variables completed during pre-test week -1 and 0 (*Lahti et al., 2020*). The sprint with the best 30-m time was used for kinematic and spatiotemporal variables. MDC was derived using Typical Error (TE) • 1.96 $\sqrt{2}$, and MDC% was defined as (MDC/$\bar{X}$) • 100. Test-retest reliability for each variable analyzed was assessed by intraclass correlation coefficient ($ICC_{3,1}$), coefficient of variation (CV%), TE with 95% confidence intervals, and MDC, using Hopkins spreadsheet (*Hopkins, 2017*). ICCs were defined as poor (ICC <0.40), fair (0.40 $\leq$ ICC <0.60), good (0.60 $\leq$ ICC <0.75), and excellent (0.75 $\leq$ ICC $\leq$ 1.00). Alpha was set at $p < 0.05$. Descriptive data are presented as mean $\pm$ standard deviation (SD).

# RESULTS

A total of four subjects could not complete the required pre-post measurements. Due to sustaining a flu, one athlete within the HS60% group could not perform final testing, making a total of nine out of 10 subjects completing the protocol. Due to injuries, three subjects in the control group could not participate in the post testing, making a total of 10 subjects measured. Furthermore, although participating in the sprint performance measurements, there was one camera malfunction during the HS50% group post-testing, leading to a loss of pre-post kinematics of one subject.

Out of 15 possible sessions, within the 9-week window the HS60% completed an average of 10.6 (95% CI [9.57–11.54]), while HS50% completed an average of 10.3 (95% CI [9.30–11.37]). For HS60%, this corresponded to a resisted sprint volume of 38.2 (95% CI [35.5–40.9]) and for HS50% 37.4 (95% CI [34.2–40.7]), $p = 0.72$.

## Group Characteristics at Baseline
All variables were normally distributed. For the final sample completing the study, baseline population variance was not significantly different for any variables, including age, height, mass, kinetic and kinematic variables ($p > 0.09$), with all split-times being highly similar (Table 1, $p > 0.55$).

## Reliability
All reliability statistical values can be found in supporting information (Tables S1–S8), including MDC%, TE, CV% and ICC. For the sprint FV-profile and performance variables, within and between session ICC ranged from good to excellent (0.60 –0.98, 95% CI [−0.09– 0.99]), except for sprint FV-profile slope and mean RF on 10-m, with RF on 10-m showing

**Table 1** **Results for sprint split-times.**

| Kinematic variables MAX | MDC (%) | Group | Pre (SD) | Post (SD) | % Δ (95%CI) | *P*-value (post-hoc), ES | Between-group statistics |
|---|---|---|---|---|---|---|---|
| | | | | | **Within-group statistics** | | |
| 5-m split time (s)[a] | 0.06 (4.00) | HS60% | 1.39 (0.05) | 1.35 (0.04) | −2.54 (−3.56; −1.52) | $p = 0.05$, ES: −0.74 | |
| | | HS50% | 1.39 (0.04) | 1.34 (0.04)** | −3.14 (−5.63; −0.65) | $p = 0.005$**, ES: −1.04 | NS |
| | | CON | 1.38 (0.04) | 1.36 (0.04) | −0.90 (−2.17; 0.88) | $p = 1.00$, ES: −0.33 | |
| 10-m split time (s)[a,b] | 0.06 (2.78) | HS60% | 2.15 (0.08) | 2.09 (0.06)** | −3.05 (−4.07; −2.03) | $p = 0.001$**, ES: −0.96 | HS50% >CON, $p = 0.03$*, ES: 1.03 |
| | | HS50% | 2.14 (0.06) | 2.07 (0.06)** | −3.37 (−5.29; −1.46) | $p < 0.001$**, ES: −1.25 | |
| | | CON | 2.12 (0.06) | 2.10 (0.04) | −0.87 (−1.95; −0.52) | $p = 0.76$, ES: −0.37 | |
| 20-m split time (s)** | 0.06 (1.71) | HS60% | 3.45 (0.12) | 3.36 (0.10)** | −2.45 (−3.37; −1.54) | $p = 0.008$**, ES: −0.77 | |
| | | HS50% | 3.43 (0.08) | 3.32 (0.10)** | −3.07 (−4.64; −1.51) | $p < 0.001$**, ES: −1.15 | NS |
| | | CON | 3.41 (0.09) | 3.37 (0.08) | −1.10 (−2.22; −0.03) | $p = 0.44$, ES: −0.47 | |
| 30-m split time (s)[a] | 0.07 (1.50) | HS60% | 4.65 (0.17) | 4.56 (0.14)* | −2.04 (−3.03; −1.06) | $p = 0.021$*, ES: −0.62 | |
| | | HS50% | 4.62 (0.10) | 4.49 (0.12)** | −2.89 (−4.15; −1.64) | $p < 0.001$, ES: −1.18 | NS |
| | | CON | 4.62 (0.12) | 4.56 (0.11) | −1.23 (−2.47; −0.26) | $p = 0.33$, ES: −0.48 | |

**Notes.**

HS, Heavy sled; CON, Control; s, seconds; Hz, Hertz; ES, Effect size (Small: 0.2–0.59, Moderate: 0.60–1.19, Large 1.19 >); SD, Standard deviation; Δ, alpha (change pre post); NS, Nonsignificant.

[a]Significant main effect of time.

[b]Significant group × time interaction effect.

*Significant post-hoc difference pre- to post-intervention ($p < .05$).

**($p < 0.01$).
poor between session reliability (0.23, 95% CI [−0.57–0.81]), and FV-profile slope showing fair reliability (0.49, 95% CI [−0.33–0.89]). For the reliability of the digitization process (within sprint spatiotemporal and kinematic variables), ICC was excellent (0.83 –0.99, 95% CI [0.38–0.99]). For the within and between session spatiotemporal and kinematic variables, ICC ranged from fair to excellent (0.41 –0.99, 95% CI [0.03–0.99]), except for maximal velocity contact time, showing poor within-session reliability (0.34, CI: −0.37; 0.80).

## Between and within group statistics
### Body mass
No significant differences were found at baseline and pre and post for BM in the 3 groups ($p > 0.05$).

### Sprint Split-times
All descriptive and inferential statistics for sprint performance can be found in Table 1 and visualized in Fig. 2. All split-times showed significant main effects for time ($p < 0.05$). Post-hoc analyses revealed significant improvements in both HS60% and HS50% for 10-m (HS60%, $p = 0.001$, $d = -0.96$; HS50%, $p < 0.001$, $d = -1.25$), 20-m (HS60%, $p = 0.008$, $d = -0.77$; HS50%, $p < 0.001$, $d = -1.15$), and 30-m split-times (HS60%, $p = 0.02$, $d = -0.62$; HS50%, $p < 0.001$, $d = -1.18$) after controlling for baseline performance. HS50% was the only group to significantly improve 5-m split-time ($p = 0.005$, $d = -1.07$), although a trend was present for HS60% ($p = 0.05$, $d = -0.74$). However, only 0–10-m, 0–20-m, and 0–30-m split time improvements surpassed the between-session minimal detectable change threshold (Fig. 2). This means that the changes in 5-m split-times could be due to normal weekly fluctuations in performance combined with measurement error. A group × time interaction effect was observed for 10-m split-time ($F(2, 24) = 4.031$, $p = 0.031$). Post-hoc analysis revealed that 10-m split-time improved significantly more in HS50% compared to CON over the study period ($p = 0.03$, $d = 1.03$).

### Sprint Force-Velocity profile variables
All within- and between-group statistics for mechanical variables can be found in Table 2. Correlations between mechanical variables can be found in Fig. 3. All mechanical variables showed significant main effects for time ($p < 0.05$). Post-hoc analyses revealed significant improvements in both HS60% and HS50% for F0 (HS60%, $p = 0.02$, $d = 1.00$; HS50%, $p = 0.002$, $d = 1.04$), Mean RF on 10-m (HS60%, $p = 0.013$, $d = 0.80$; HS50%, $p < 0.001$, $d = 1.14$), and Pmax (HS60%, $p = 0.011$, $d = 0.84$; HS50% , $p < 0.001$, $d = 1.18$) after controlling for baseline values. However, the F0 changes (HS60%: 7.83%, HS50%: 9.23%) were under the between-session minimal detectable change threshold (9.53%). RFmax improved significantly in all groups (HS60%, $p = 0.011$, $d = 1.25$; HS50%, $p = 0.001$, $d = 1.01$; CON, $p = 0.041$, $d = 0.55$). There was a significant improvement in HS50% for v0 ($\Delta$ 3.08%, $p = 0.04$, $d = 0.78$), however, the result remained under the between session minimal detectable change threshold (3.13%). No other within-group significant changes were observed ($p > 0.05$). A group × time interaction effect was observed for Pmax ($F(2, 24) = 4.055$, $p = 0.030$), and a trend for F0 ($F(2, 24) = 2.778$, $p = 0.082$).
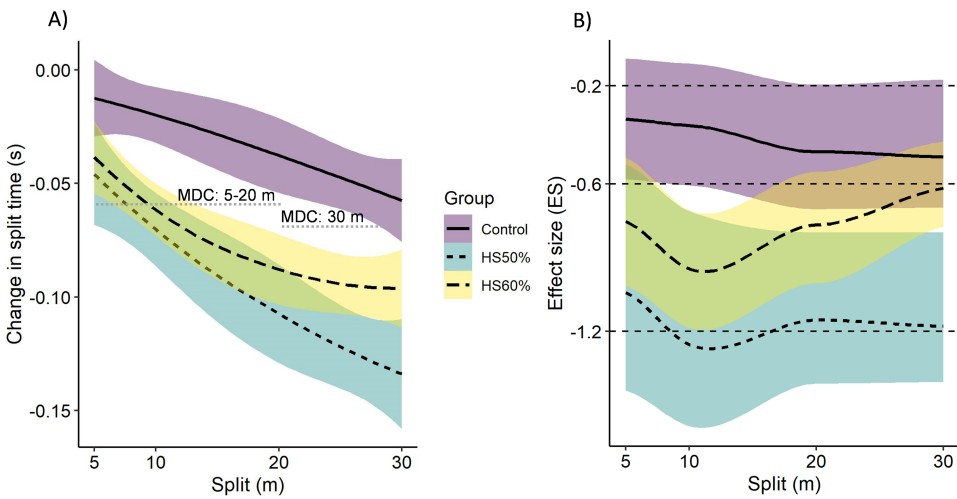

**Figure 2** **Sprint split-time changes.** Raw Changes in split time performance with MDC thresholds (A) and their corresponding effect sizes within each group with ES thresholds (B). The lines between the four split-time measurements (0-5, 0-10, 0-20, 0-30) have been smoothed. The error ribbons represent standard error via bias corrected and accelerated bootsrapping at 0.68 confidence intervals, corresponding to +/- 1 standard deviation. HS: Heavy sled, CON: control group, MDC: Minimal detectable change.

Post-hoc analysis revealed that Pmax improved significantly more in HS50% compared to CON over the study period ($p = 0.03$, $d = 1.16$). No other between-group differences were observed.

### Sprint kinematic and spatiotemporal variables

*Cross-sectional analysis of immediate effects of sled on early acceleration.* All significant results for immediate effects of sled are visualized in Fig. 4. All descriptive and inferential statistics can be found in Table 3. Due to timetable issues, eight out of nine subjects were available for kinematic filming of the sled from the HS60% group and six out of nine from the HS50% group.

Between-group t-tests showed no differences ($p > 0.05$). Within group $t$-test comparisons with Benjamini–Hochberg corrections showed that the provided resistance from the sled led to significant changes in both spatiotemporal and kinematic variables. All spatiotemporal variables changed significantly in the HS60% group, with increased contact time ($p = 0.003$, $d = 2.10$), step rate ($p = 0.004$, $d = -1.90$), and step length ($p = 0.008$: $d = -1.58$). Both sled loads significantly decreased touchdown CM distance (HS60%: $p = 0.003$, $d = 1.99$; HS50%: $p = 0.003$, $d = 3.50$) and CM angle at touchdown (HS60%: $p = 0.005$, $d = -2.30$, HS50%: $p = 0.005$, $d = -3.00$), corresponding to taking steps further behind center of mass. No other variables reached significance ($p > 0.05$).

*Pre-Post intervention changes in kinematic and spatiotemporal variables.* All descriptive and inferential statistics for sprint technique can be found in Table 4 and visualized in Fig. 5. In early acceleration, there was a main effect for time in step length, contralateral hip angle at toe-off, and contralateral hip angle at touchdown. At maximal velocity, there was

**Table 2  Results for sprint mechanical variables.**

| Variable | MDC % | Group | Pre (SD) | Post (SD) | % Δ (95% CI) | P-value (post-hoc), ES | Between-group statistics |
|---|---|---|---|---|---|---|---|
| | | | Within-group statistics | | | | Between-group statistics |
| $F0$ (N.kg$^{-1}$)[a] | 0.68 (9.53) | HS60% | 7.23 (0.63) | 7.77 (0.42)* | 7.83 (4.16; 11.5) | $p = 0.018$*, ES: 1.00 | NS |
| | | HS50% | 7.27 (0.59) | 7.91 (0.65)** | 9.23 (3.58; 14.9) | $p = 0.002$**, ES: 1.04 | |
| | | CON | 7.43 (0.50) | 7.58 (0.45) | 1.89 (−1.60; 5.39) | $p = 1.00$, ES: 0.30 | |
| RFmax (%)[a] | 1.64 | HS60% | 47.9 (2.57) | 50.8 (1.88)* | 6.03 (4.01; 8.03) | $p = 0.011$*, ES: 1.25 | NS |
| | | HS50% | 47.9 (3.51) | 51.2 (2.91)** | 7.12 (2.59; 11.7) | $p = 0.001$**, ES: 1.01 | |
| | | CON | 50.1 (2.39) | 51.6 (2.58)* | 3.00 (0.42; 5.58) | $p = 0.041$, ES: 0.55 | |
| Mean RF on 10-m (%)[a] | 4.99 | HS60% | 27.7 (1.71) | 28.9 (1.42)* | 4.70 (2.83; 6.58) | $p = 0.013$*, ES: 0.80 | NS |
| | | HS50% | 27.9 (1.59) | 29.8 (1,61)** | 6.58 (4.00; 9.17) | $p < 0.001$, ES: 1.14 | |
| | | CON | 28.6 (1.61) | 29.3 (1.36) | 3.20 (0.95; 5.45) | $p = 0.05$, ES: 0.65 | |
| Pmax (W.kg$^{-1}$)[a,b] | 1.10 (6.97) | HS60% | 16.0 (1.66) | 17.3 (1.35)* | 8.36 (5.11; 11.6) | $p = 0.011$*, ES: 0.84 | HS50% vs CON: $p = 0.03$*, ES: 1.16 |
| | | HS50% | 16.2 (1.31) | 18.1 (1.82)** | 11.64 (6.40; 16.9) | $p < 0.001$, ES: 1.18 | |
| | | CON | 16.5 (1.27) | 17.0 (1.08) | 4.05 (0.94; 7.15) | $p = 0.70$, ES:: 0.49 | |
| $v0$ (m.s$^{-1}$)[a] | 0.28 (3.13) | HS60% | 8.93 (0.51) | 9.08 (0.39) | 1.79 (−0.21; 3.78) | $p = 1.00$, ES: 0.32 | NS |
| | | HS50% | 9.03 (0.36) | 9.31 (0.33)* | 3.08 (1.44; 4.72) | $p = 0.044$*, ES: 0.78 | |
| | | CON | 8.96 (0.36) | 9.10 (0.42) | 2.04 (−0.45; 4.54) | $p = 1.00$, ES: 0.34 | |
| Sprint FV-profile (-$F0/v0$) [a] | 0.06 (7.37) | HS60% | −0.81 (0.08) | −0.86 (0.05) | 6.07 (1.54; 10.62) | $p = 0.29$, ES: −0.67 | NS |
| | | HS50% | −0.81 (0.08) | −0.85 (0.06) | 6.11 (−0.30; 12.5) | $p = 0.57$, ES: −0.60 | |
| | | CON | −0.83 (0.07) | −0.83 (0.07) | 0.12 (−5.31; 5.56) | $p = 1.00$, ES: −0.06 | |

Notes.

$F_0$, Heavy sled; CON, Control; s, seconds; Hz, Hertz; ES, Effect size (Small: 0.2–0.59, Moderate: 0.60–1.19, Large 1.19 >); SD, Standard deviation; Δ, alpha (change pre post); NS, Nonsignificant.

[a]Significant main effect of time.

[b]Significant group ×time interaction effect.

*Significant post-hoc difference pre- to post-intervention ($p < .05$).

**($p < 0.01$).

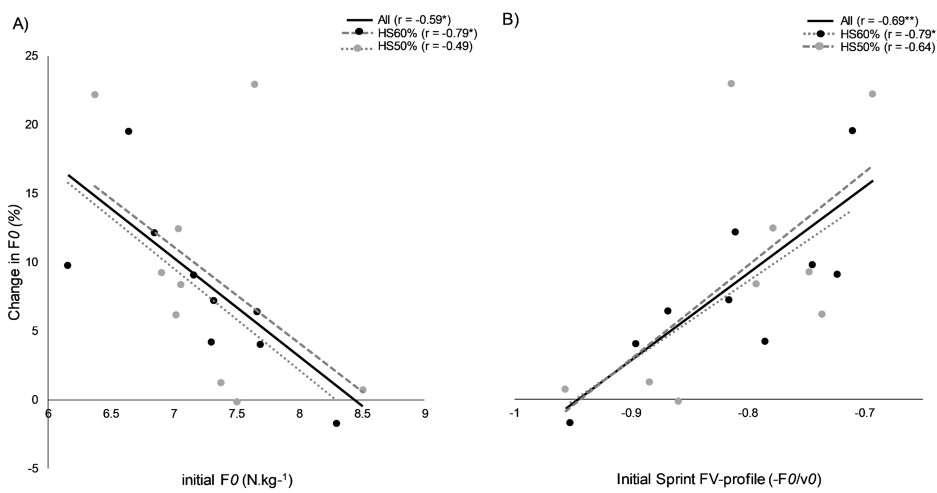

**Figure 3 Mechanical variable correlations.** Correlation coefficients between initial values in (A) maximal theoretical horizontal force (F0) production, (B) initial Sprint FV-profile (-F0/v0), and respective changes post intervention. HS: Heavy sled, CON: control group, *: $p < 0.05$, **: $p < 0.01$.

a main effect for time in step rate, trunk angle at toe-off, hip contralateral angle at toe-off, and CM angle at toe-off. Post-hoc analyses revealed a significant decrease in both HS60% and CON for contralateral hip angle at touchdown during early acceleration (HS60%: Δ -4.01%, $p = 0.004$, $d = -0.80$; CON: Δ -3.13%, $p = 0.006$, $d = -0.80$) after controlling for baseline values. However, the result remained under the between session minimal detectable change threshold (5.85%). All other within-group comparisons did not reach significance ($p > 0.05$).

No interaction effects were found for pre and post sprint kinematic och spatiotemporal variables for both early acceleration and upright sprinting ($p > 0.05$).

## DISCUSSION

The main results of this study were that, although both heavy load conditions (50% and 60% velocity decrement) improved sprint performance in soccer players, the HS50% was the only group showing changes in sprint parameters that were significantly different from CON. A clear favoring towards improvements in early acceleration performance and sprint kinetics were present in both HS50% and HS60% groups, showing moderate to large effect size differences compared to CON. Furthermore, although both loads produced significant immediate changes in early acceleration at toe-off and touchdown, no long-term changes on early acceleration and upright sprint technique were observed that surpassed the minimal detectable change. These results suggest that heavy resisted sprinting can be successfully integrated in a professional soccer setting, potentially preferably with resistance associated to a ~50% drop in maximal running velocity compared to ~60%.

Our initial hypothesis was partly met, with heavy resisted sprinting leading to improved early acceleration sprint performance. It is important to mention that the reported 5-m within-group improvements fell under the minimal detectable change threshold and, thus,

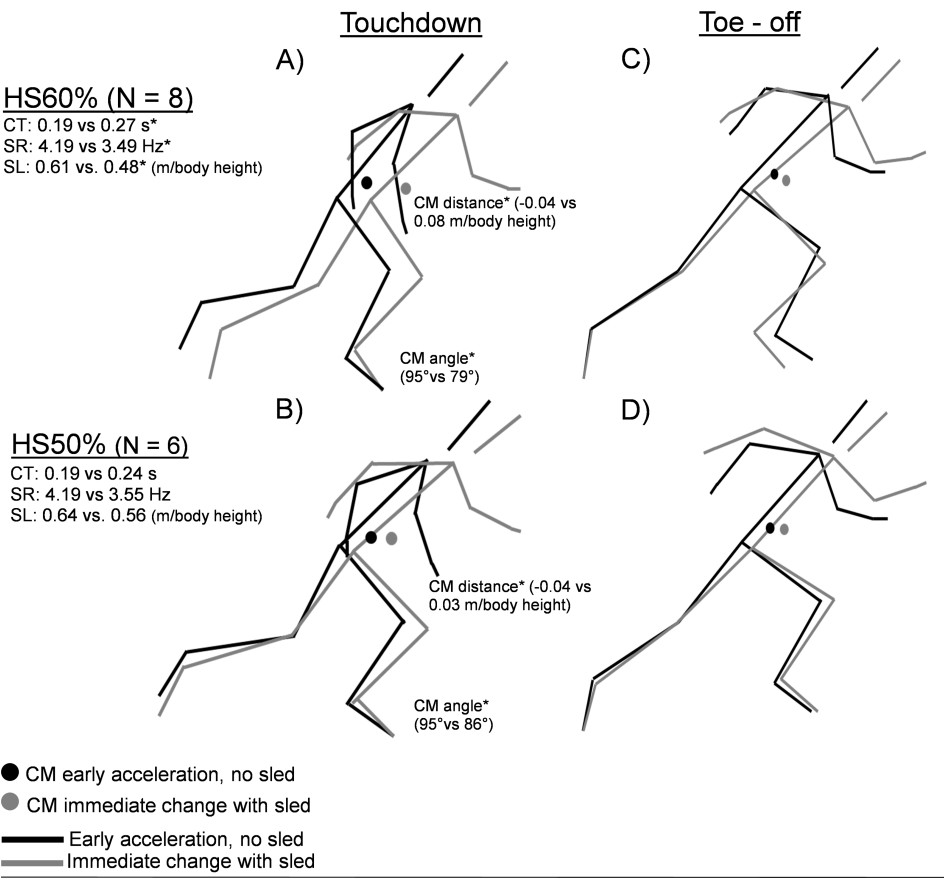

Touchdown  Toe - off

A)

**HS60% (N = 8)**
CT: 0.19 vs 0.27 s*
SR: 4.19 vs 3.49 Hz*
SL: 0.61 vs. 0.48* (m/body height)

CM distance* (-0.04 vs 0.08 m/body height)

CM angle* (95° vs 79°)

C)

B)

**HS50% (N = 6)**
CT: 0.19 vs 0.24 s
SR: 4.19 vs 3.55 Hz
SL: 0.64 vs. 0.56 (m/body height)

CM distance* (-0.04 vs 0.03 m/body height)

CM angle* (95°vs 86°)

D)

● CM early acceleration, no sled
● CM immediate change with sled
▬ Early acceleration, no sled
▬ Immediate change with sled

**Figure 4  Sprint kinematic and spatiotemporal changes, immediate effects of sled.** Immediate kinematic and spatiotemporal differences between early acceleration (black) and sled sprinting (gray). Touchdown (A, B) and toe –off (C, D) within HS60% and HS50% groups. HS: Heavy sled, CT: Contact time, SR: Step Rate, SL: Step Length relative to body height, CM: Center of Mass, IPSI: Ipsilateral (ground contact leg), m: meter, *: $p < 0.05$. No group differences were found ($p < 0.05$).

still could be interpreted as remaining within the measurement error thresholds (Fig. 2). This is a logical result based on previous literature on 5-m split time measurements (*Bezodis, Salo & Trewartha, 2012*). However, we expected to see differences between the heavy loads in improving specific parts of early acceleration sprint performance. Specifically, we expected the HS60% group to mostly improve the 0–5-m split-times, whereas the HS50% group would mostly improve the 0–10-m split times. This is because the first steps of acceleration are considered to be more dependent on maximal force capacity, with its importance reducing with increasing velocity (*Kawamori, Nosaka & Newton, 2013*; *Cottle, Carlson & Lawrence, 2014*). Hence the larger load was thought to provide a higher transfer in this area. However, both heavy loads had similar effects on early acceleration performance (Fig. 2). Although the HS50% group was the only group to reach significantly lower split-times compared to CON and had a large effect size (0–10-m split-time). Furthermore, Fig. 2 shows trends towards HS50% providing a broader stimulus across the entire acceleration phase. Future studies should verify how reproducible this
adaptation signature is. The most evident reasons for the lack of differences in loads can be a combination of a too small difference in loading parameters and that the total training volume was possibly not high enough.

The underlying kinetic reasons to the performance improvements were also of interest in this study. Therefore, we analyzed the ratio of forces at the first step and over the first 10-m (RFmax and mean RF on 10-m). However, caution should be considered within the interpretation of mean RF on 10-m, showing poor between-session reliability within this population. The analysis showed that when considering initial values, there was a lack of clear difference in effect size between RFmax and $F_0$ compared to the control group (both moderate effects). Therefore, improvements in both their maximal ground reaction force capacity and their capability to orient this force more horizontally may have contributed to improved sprint performance. However, Pmax was the only biomechanical variable to show significant improvements compared to CON, specifically in the HS50% group. As external maximal power is produced at approximately 50% of maximal velocity in a maximal acceleration sprint (*Cross et al., 2017*), it makes sense that Pmax was maximized in the HS50% group. Therefore, the ability to produce higher forces at higher velocities (i.e., maximal mechanical power), seemed to be the main driver for the improved sprint performance.

The most important aim of improving sprint performance was met, an essential part in preparing soccer athletes for the season (*Haugen et al., 2014*; *Cometti et al., 2001*). This contradicted previous literature with similar loading parameters. Specifically, the main methodological strengths of this study compared to previous literature were that the present groups were evenly divided based on their initial sprint performance, training was done mostly 1–2 per week instead of once, and tapering was completed (*Pareja-Blanco, Asián-Clemente & SáezdeVillarreal, 2019*; *Cross et al., 2018*). Furthermore, in the study by *Pareja-Blanco, Asián-Clemente & SáezdeVillarreal (2019)* loads were not standardized and individualized to a specific velocity decrement, but rather to body mass (80% of BM). Therefore, one conclusion is that if a time slot of roughly 20 min is accepted for velocity-based resisted sprint training within field practice conditions 1–2 per week, it will likely be beneficial, assuming the athlete has been assessed for lacking early acceleration capacity (Fig. 4). However, our study did not have a group completing non-resisted sprint training, only a control group completing sport-specific training. Therefore, we do not know if just the mere systematic focus on early acceleration, regardless of load, is enough. Measuring a force-velocity and load-velocity profile for everyone might be an issue for some as there may be time constraints and lack of access to technology. However, this can be done relatively quickly and at a low cost with the help of accurate apps (*Romero-Franco et al., 2017*), while saving some time with a shorter load-velocity protocol (3 loads: 0, 25 and 75% of BM is sufficient to obtain the linear individual load-velocity profile, see Fig. 2 in *Cross et al. (2018)*), although this still needs to be validated.

Our second hypothesis was that both loads would improve early acceleration toe-off CM distance (more triple extension of the body) and CM angle (increased forward body lean). The results showed no changes in the kinematics or any other variables in early acceleration, which is in contrast to previous light load literature showing slight increases in trunk lean

**Table 3  Results for kinematic variables from immediate effects on early acceleration of sled loads.**

| Variable | Group | Toe-off without sled | Toe-off with sled | % Δ ± CI 95% | Within group Statistics (*P*-value, ES) | Touchdown without sled | Touchdown with sled | % Δ ± CI 95% | Within group Statistics (*P*-value, ES) |
|---|---|---|---|---|---|---|---|---|---|
| CM distance (m/body length) | HS60% | 0.42 (0.04) | 0.45 (0.03) | 7.74 (−0.53; 16.0) | *p* = 0.15, ES: 0.85 | −0.04 (0.03) | 0.08 (0.08) | −820 (−1670; 29.3) | *p* = 0.003*, ES: 1.99 |
| | HS50% | 0.43 (0.01) | 0.46 (0.03) | 7.18 (3.31; 11.0) | *p* = 0.03, ES: 1.34 | −0.04 (0.02) | 0.03 (0.02) | −847 (−1751; 55.9) | *p* = 0.003*, ES: 3.50 |
| CM angle (°) | HS60% | 46.8 (1.77) | 44.1 (2.21) | −5.79 (−9.90; −1.67) | *p* = 0.04, ES: −1.49 | 95.3 (4.19) | 79.8 (8.59) | −16.1 (−22.9; −11.0) | *p* = 0.005*, ES: −2.30 |
| | HS50% | 46.6 (1.22) | 44.7 (1.49) | −4.46 (7.41; −1.52) | *p* = 0.06, ES: −2.33 | 95.2 (3.30) | 86.2 (2.60) | −8.46 (−11.0; −5.97) | *p* = 0.005*, ES: −3.00 |
| Hip-angle Ipsilateral (°) | HS60% | 171 (7.61) | 173 (10.6) | 2.05 (−1.91; 6.01) | *p* = 0.41, ES: 0.10 | 101 (7.30) | 108 (20.3) | 7.67 (−9.25; 24.6) | *p* = 0.40, ES: 0.41 |
| | HS50% | 174 (2.95) | 181 (4.82) | 4.22 (1.33; 7.11) | *p* = 0.07, ES: 1.70 | 105 (8.10) | 108 (4.04) | 3.18 (−1.40; 7.78) | *p* = 0.28, ES: 0.60 |
| Hip-angle Contralateral (°) | HS60% | 85.7 (6.72) | 90.3 (7.16) | 6.01 (−3.30; 15.3) | *p* = 0.19, ES: 0.57 | 161 (8.81) | 159 (13.1) | −0.34 (−6.44; 5.76) | *p* = 0.71, ES: −0.18 |
| | HS50% | 86.7 (4.08) | 84.7 (6.09) | −3.81 (−7.58; −0.02) | *p* = 0.45, ES: −0.59 | 164 (6.59) | 164 (10.2) | 2.56 (−2.08; 7.21) | *p* = 0.91, ES: 0.00 |
| Trunk angle (°) | HS60% | 46.3 (5.20) | 42.7 (8.37) | −6.09 (−19.0; 6.82) | *p* = 0.29, ES: −0.60 | 46.8 (6.18) | 42.0 (8.11) | −7.54 (−21.2; 6.11) | *p* = 0.18, ES: −0.85 |
| | HS50% | 47.9 (2.87) | 49.4 (2.76) | 1.12 (−4.25; 6.50) | *p* = 0.31, ES: 0.33 | 49.1 (3.97) | 48.4 (2.40) | −1.77 (−6.45; 2.90) | *p* = 0.66, ES: −0.19 |
| **Spatiotemporal variables** | **Group** | **Early acceleration, no sled** | **Early acceleration, with sled** | | | | | **% Δ ± CI 95%** | **Within group Statistics (*P*-value, ES)** |
| Contact time (s) | HS60% | 0.191 (0.02) | 0.274 (0.05) | | | | | 40.0 (24.5; 55.4) | *p* = 0.003*, ES: 2.10 |
| | HS50% | 0.193 (0.01) | 0.240 (0.04) | | | | | 28.2 (13.3; 43.1) | *p* = 0.03, ES: 1.71 |
| Step Rate (Hz) | HS60% | 4.19 (0.20) | 3.49 (0.51) | | | | | −16.5 (−23.3; −9.70) | *p* = 0.004*, ES: −1.90 |
| | HS50% | 4.19 (0.17) | 3.55 (0.41) | | | | | −14.8 (−23.6; −6.12) | *p* = 0.041, ES: −2.09 |
| Step Length (m/body length) | HS60% | 0.61 (0.06) | 0.48 (0.10) | | | | | −21.9 (−32.3; −11.5) | *p* = 0.008*, ES: −1.58 |
| | HS50% | 0.64 (0.04) | 0.56 (0.04) | | | | | −11.3 (−16.7; −5.97) | *p* = 0.02, ES: −.2.00 |

**Notes.**

HS, Heavy sled; CON, Control; TO, Toe-off; TD, Touchdown; CM, Center of Mass; m, meter; s, seconds; Hz, Hertz; ES, Effect size (Small: 0.2–0.59, Moderate: 0.60–1.19, Large 1.19 >); SD, Standard deviation; Δ, alpha (change pre post).

*Significant difference after controlling for multiple comparisons using the Benjamini–Hochberg procedure.

Lahti et al. (2020), *PeerJ*, DOI 10.7717/peerj.10507

*Peer*J

**Table 4  Results for kinematic and spatiotemporal variables in early acceleration (ACC) and upright sprinting (MAX).**

| Kinematic variables ACC | MDC (%) Toe-off | MDC (%) Touchdown | Group | ACC Toe-off pre (SD) | ACC Toe-off post (SD) | % Δ (95% CI) | P-value (post-hoc), ES | ACC Touchdown pre (SD) | ACC Touchdown post (SD) | % Δ (95% CI) | P-value (post-hoc), ES |
|---|---|---|---|---|---|---|---|---|---|---|---|
| | | | | | | | | Within-group statistics | | | |
| CM distance m/body length | 0.04 (4.76) | 0.01 (−55.7) | HS60% | 0.42 (0.03) | 0.42 (0.04) | −0.01 (−1.56; 1.36) | p = 1.00, ES: −0.01 | −0.04 (0.03) | −0.03 (0.03) | 39.0 (−79.2; 157) | p = 1.00, ES: 0.39 |
| | | | HS50% | 0.43 (0.01) | 0.43 (0.01) | 0.16 (−1.22; 1.56) | p = 1.00, ES: 0.04 | −0.04 (0.02) | −0.02 (0.03) | 35.0 (−420; 490) | p = 0.55, ES: 0.70 |
| | | | CON | 0.43 (0.02) | 0.44 (0.01) | 1.04 (−0.82; 2.10) | p = 1.00, ES: 0.16 | −0.03 (0.03) | −0.03 (0.02) | 156 (−227; 540) | p = 1.00, ES: 0.00 |
| CM angle (°)[a] Relative to horizontal | 1.29 (2.75) | 2.19 (2.36) | HS60% | 46.8 (1.77) | 47.4 (1.38) | 1.32 (−0.59; 3.23) | p = 1.00, ES: 0.36 | 95.3 (4.19) | 93.7 (3.37) | −1.63 (−3.02; −0.25) | p = 1,00, ES: −0.42 |
| | | | HS50% | 46.6 (1.22) | 46.8 (1.08) | 0.46 (−0.64; 1.57) | p = 1.00, ES: 0.17 | 95.2 (3.30) | 92.6 (4.18) | −2.66 (−6.16; 0.82) | p = 0.46, ES: −0.69 |
| | | | CON | 47.7 (1.97) | 47.5 (1.24) | 0.45 (−0.81; 1.71) | p = 1.00, ES: 0.11 | 93.7 (4.99) | 93.3 (3.13) | −0.32 (−2.36; 1.72) | p = 1.00, ES: −0.10 |
| Hip-angle Ipsilateral (°) 180° = full EXT | 6.31 (3.73) | 10.7 (10.2) | HS60% | 171 (7.61) | 169 (6.72) | −1.19 (−3.07; 0.68) | p = 0.72, ES: −0.30 | 101 (7.30) | 103 (5.28) | 1.94 (−2.25; 6.14) | p = 1.00, ES: 0.26 |
| | | | HS50% | 174 (2.95) | 175 (2.69) | 0.12 (−1.59; 1.82) | p = 1.00, ES: 0.05 | 104 (8.10) | 105 (6.14) | 0.74 (−3.27; 4.75) | p = 1.00, ES: 0.07 |
| | | | CON | 170 (5.28) | 171 (3.18) | 0.41 (−0.51; 1.33) | p = 1.00, ES: 0.14 | 103 (8.73) | 103 (5.95) | 1.22 (−2.01; 4.44) | p = 1.00, ES: 0.12 |
| Hip-angle Contralateral (°)[a] 180° = full EXT | 5.97 (7.11) | 9.12 (5.85) | HS60% | 85.7 (6.72) | 82.8 (3.98) | −3.03 (−5.91; −0.15) | p = 0.33, ES: −0.51 | 161 (8.81) | 154 (7.49) | −4.01 (−5.97; −2.05) | p = 0.004**, ES: −0.80 |
| | | | HS50% | 86.7 (4.08) | 85.6 (5.74) | −1.25 (−4.62; 2.10) | p = 1.00, ES: −0.22 | 164 (6.59) | 162 (4.87) | −1.57 (−4.68; 1.56) | p = 1.00, ES: −0.48 |
| | | | CON | 85.1 (8.98) | 84.6 (8.04) | −0.47 (−2.39; 1.46) | p = 1.00, ES: −0.06 | 159 (7.18) | 155 (5.36) | −3.13 (−4.65; −1.61) | p = 0.006**, ES: −0.80 |
| Trunk angle (°) Relative to horizontal | 4.97 (10.8) | 6.62 (14.2) | HS60% | 46.3 (5.20) | 45.3 (3.03) | −1.48 (−6.44; 3.47) | p = 1.00, ES: −0.23 | 46.8 (6.18) | 45.9 (2.59) | −0.73 (−7.25; 5.79) | p = 1.00, ES: −0.18 |
| | | | HS50% | 47.9 (2.87) | 48.6 (3.77) | 1.44 (−2.54; 5.41) | p = 1.00, ES: 0.20 | 49.1 (3.97) | 48.8 (4.25) | −0.39 (−4.50; 4.21) | p = 1.00, ES: −0.07 |
| | | | CON | 46.5 (5.29) | 46.6 (4.29) | 0.59 (−2.10; 3.28) | p = 1.00, ES: 0.03 | 47.3 (5.50) | 46.0 (4.24) | −2.26 (−6.25; 1.73) | p = 1.00, ES: −0.26 |

| Spatiotemporal variables ACC | MDC (%) | Group | Pre (SD) | Post (SD) | % Δ (95% CI) | P-value (post-hoc), ES |
|---|---|---|---|---|---|---|
| Contact time (s) | 0.02 (9.32) | HS60% | 0.19 (0.02) | 0.18 (0.02) | −5.48 (−9.12; −1.83) | p = 1.00, ES: 0.56 |
| | | HS50% | 0.19 (0.01) | 0.19 (0.03) | −0.97 (−13.0; 11.01) | p = 1.00, ES: −0.12 |
| | | CON | 0.19 (0.01) | 0.18 (0.01) | −2.34 (−6.50; 1.82) | p = 1.00, ES: −0.34 |
| Step Rate (Hz) | 0.25 (5.71) | HS60% | 4.19 (0.20) | 4.32 (0.29) | 3.25 (−0.56; 7.07) | p = 1.00, ES: 0.54 |
| | | HS50% | 4.19 (0.17) | 4.36 (0.41) | 4.45 (−3.09; 12.0) | p = 1.00, ES: 0.56 |
| | | CON | 4.27 (0.26) | 4.28 (0.33) | 0.54 (−2.61; 3.69) | p = 1.00, ES: 0.08 |

Lahti et al. (2020), *PeerJ*, DOI 10.7717/peerj.10507

## Table 4 (*continued*)

| Kinematic variables ACC | MDC (%) Toe-off | MDC (%) Touchdown | Group | ACC Toe-off pre (SD) | ACC Toe-off post (SD) | % Δ (95% CI) | P-value (post-hoc), ES | ACC Touchdown pre (SD) | ACC Touchdown post (SD) | % Δ (95% CI) | P-value (post-hoc), ES |
|---|---|---|---|---|---|---|---|---|---|---|---|
| | | | HS60% | 0.61 (0.06) | 0.62 (0.06) | 1.52 (−3.21; 6.24) | p = 1.00, ES: 0.13 | | | | |
| Step Length (m/body length)[a] | 0.05(4.89) | | HS50% | 0.64 (0.03) | 0.64 (0.04) | 0.15 (−2.96; 3.26) | p = 1.00, ES: −0.50 | | | | |
| | | | CON | 0.62 (0.05) | 0.65 (0.05) | 5.38 (1.12; 9.64) | p = 0.23, ES: 0.60 | | | | |

| Kinematic variables MAX | MDC (%) Toe-off | MDC (%) Touch-down | Group | MAX Toe-off pre (SD) | MAX Toe-off post (SD) | % Δ (95% CI) | P-value (post-hoc), ES | MAX Touchdown pre (SD) | MAX Touchdown post (SD) | % Δ (95% CI) | P-value (post-hoc), ES |
|---|---|---|---|---|---|---|---|---|---|---|---|
| CM distance to toe m/-body length | 0.05 (8.27) | 0.04 (−12.1) | HS60% | 0.35 (0.01) | 0.34 (0.01) | −2.09 (−3.76; −0.41) | p = 1.00, ES: −0.48 | −0.23 (0.02) | −0.21 (0.02) | −5.84 (−10.9; −0.83) | p = 0.63, ES: 0.71 |
| | | | HS50% | 0.34 (0.02) | 0.36 (0.03) | 3.67 (−1.51; 8.85) | p = 0.63, ES: 0.44 | −0.22 (0.02) | −0.21 (0.01) | −2.81 (−6.77; 1.16) | p = 1.00, ES: 0.44 |
| | | | CON | 0.33 (0.02) | 0.33 (0.02) | −0.19 (−1.57; 1.19) | p = 1.00, ES: −0.02 | −0.21 (0.02) | −0.21 (0.02) | −1.11 (−4.75; 2.53) | p = 1.00, ES: 0.09 |
| CM angle (°)[a] | 2.21 (3.87) | 2.94 (2.64) | HS60% | 56.6 (2.13) | 57.1 (1.87) | 0.95 (0.19; 1.71) | p = 1.00, ES: 0.26 | 114 (2.11) | 112 (2.11) | −1.23 (−2.27; −0.20) | p = 0.50, ES: −0.67 |
| | | | HS50% | 57.6 (2.77) | 56.1 (2.63) | −2.48 (0.19; 0.44) | p = 0.55, ES: −0.54 | 112 (1.64) | 112 (2.01) | −0.44 (−1.16; 0.28) | p = 1.00, ES: −0.27 |
| | | | CON | 56.4 (2.38) | 57.7 (2.17) | 2.40 (0.77; 4.03) | p = 0.32, ES: 0.58 | 112 (2.37) | 112 (2.49) | 0.03 (−0.83; 0.90) | p = 1.00, ES: 0.01 |
| Hip-angle Ipsilateral (°) | 3.56 (1.77) | 5.40 (4.03) | HS60% | 201 (4.46) | 201 (5.14) | 0.13 (−0.99; 1.25) | p = 1.00, ES: 0.05 | 134 (6.15) | 136 (5.40) | 1.69 (−0.18; 3.56) | p = 1.00, ES: 0.38 |
| | | | HS50% | 202 (5.38) | 202 (4.22) | −0.34 (−1.51; 0.82) | p = 1.00, ES: −0.15 | 141 (14.3) | 140 (3.81) | −0.39 (−2.46; 1.67) | p = 1.00, ES: −0.04 |
| | | | CON | 202 (5.84) | 201 (5.79) | −0.27 (−0.87; 0.32) | p = 1.00, ES: −0.10 | 135 (5.57) | 136 (5.82) | 0.41 (−1.51; 2.33) | p = 1.00, ES: 0.08 |
| Hip-angle Contralateral (°) | 3.92 (3.67) | 6.17 (3.60) | HS60% | 105 (3.42) | 106 (4.94) | 0.52 (−1.29; 2.33) | p = 1.00, ES: 0.13 | 176 (4.69) | 173 (4.92) | −1.64 (−3.77; 0.49) | p = 1.00, ES: −0.61 |
| | | | HS50% | 107 (8.24) | 104 (4.26) | −2.08 (−5.21; 1.04) | p = 1.00, ES: −0.39 | 174 (7.85) | 172 (4.80) | −1.37 (−3.38; 0.64) | p = 1.00, ES: −0.39 |
| | | | CON | 106 (4.54) | 107 (5.79) | 1.13 (−1.17; 3.44) | p = 1.00, ES: 0.23 | 171 (11.6) | 169 (13.2) | −1.40 (−3.44; 0.64) | p = 1.00, ES: −0.19 |
| Trunk angle (°)[a] | 2.14 (2.79) | 2.40 (3.15) | HS60% | 78.7 (4.37) | 79.3 (4.36) | 0.87 (−0.74; 2.51) | p = 1.00, ES: 0.15 | 79.9 (3.92) | 80.4 (3.99) | 0.61 (−1.66; 2.89) | p = 1.00, ES: 0.11 |
| | | | HS50% | 78.9 (5.48) | 77.6 (3.48) | −1.48 (−4.23; 1.27) | p = 1.00, ES: -0.29 | 78.6 (4.43) | 78.5 (3.86) | −0.09 (−2.22; 2.03) | p = 1.00, ES: −0.03 |
| | | | CON | 78.0 (5.54) | 79.4 (3.73) | 2.03 (−1.60; 5.68) | p = 1.00, ES: 0.28 | 77.9 (4.47) | 79.0 (3.74) | 1.52 (−0.96; 4.01) | p = 1.00, ES: 0.26 |

| Spatiotemporal variables MAX | MDC (%) | Group | Pre (SD) | Post (SD) | % Δ (95% CI) | P-value (post-hoc), ES |
|---|---|---|---|---|---|---|
| Contact time (s) | 0.01 (10.9) | HS60% | 0.13 (0.01) | 0.12 (0.02) | −2.52 (−7.53 −2.48) | p = 1.00, ES: −0.32 |
| | | HS50% | 0.12 (0.01) | 0.12 (0.01) | −2.70 (−6.64 −1.23) | p = 1.00, ES: −0.51 |
| | | CON | 0.12 (0.01) | 0.12 (0.01) | 0.56 (−2.47 −3.59) | p = 1.00, ES: 0.09 |
| Step Rate (Hz)[a] | 0.30 (6.60) | HS60% | 4.30 (0.25) | 4.48 (0.19) | 4.38 (1.62 −7.14) | p = 0.12, ES: 0.82 |
| | | HS50% | 4.47 (0.12) | 4.65 (0.12) | 4.00 (1.66 −6.33) | p = 0.90, ES: 1.50 |
| | | CON | 4.50 (0.18) | 4.53 (0.28) | 0.67 (−2.82 −4.17) | p = 1.00, ES: 0.12 |
| Step Length m/body length | 0.08 (4.53) | HS60% | 1.04 (0.04) | 1.02 (0.03) | −1.39 (−3.07 −0.28) | p = 1.00, ES: −0.39 |
| | | HS50% | 1.08 (0.06) | 1.07 (0.07) | −1.37 (−3.75 −1.00) | p = 1.00, ES: −0.23 |
| | | CON | 1.03 (0.08) | 1.01 (0.06) | −1.38 (−5.26 −2.50) | p = 1.00, ES: −0.23 |

**Notes.**

HS, Heavy sled; CON, Control; TO, Toe-off; TD, Touchdown; CM, Center of Mass; m, meter; s, seconds; Hz, Hertz; ES, Effect size (Small: 0.2–0.59, Moderate: 0.60–1.19, Large 1.19 >); SD, Standard deviation; Δ, alpha (change pre post); NS, Nonsignificant.

[a]Significant main effect of time.

*Significant post-hoc difference pre- to post-intervention ($p < .05$).

**($p < 0.01$).

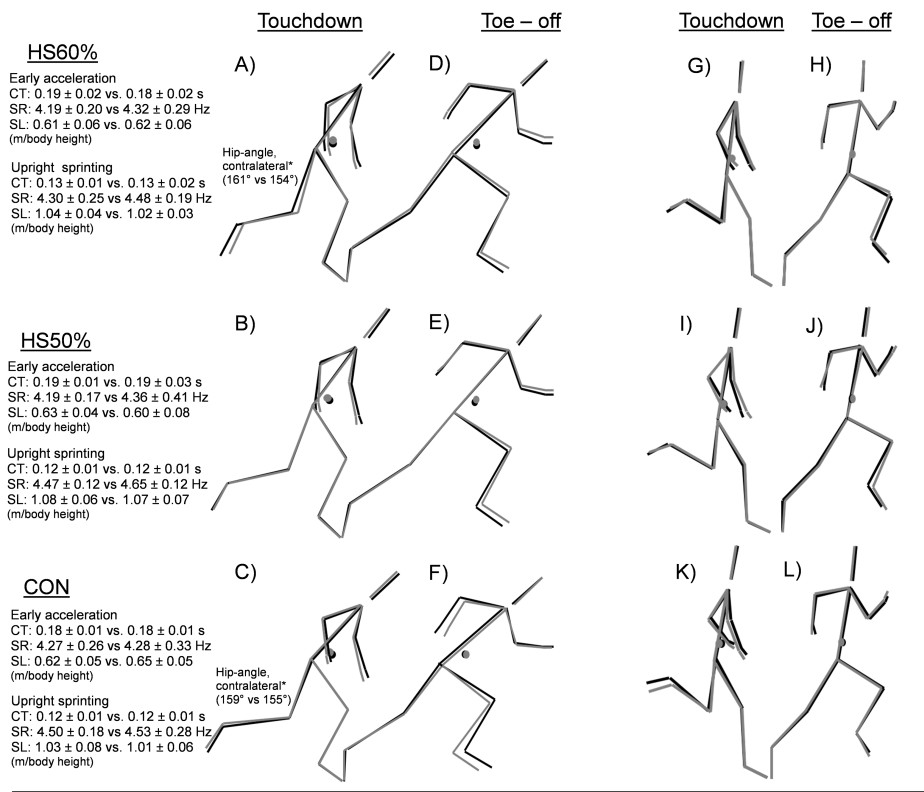

**Figure 5** **Pre-post intervention sprint kinematic changes in early acceleration and upright sprinting.** Touchdown (A, B, C, J, I, K) and toe –off (D, E, F, H, J, L) within HS60%, HS50%, and CON groups. In early acceleration, toe-off is based on the average of the first push toe-off from the sprint start and the first two steps toe-off. The touchdown is based on the first 3 steps. Upright sprinting toe-off and touchdown are analyzed from 2 steps during upright sprinting at our close to maximal velocity ($\sim$22.5 m). No kinematic variables for within and between-group comparisons reached significance. HS: Heavy sled, CT: Contact time, SR: Step rate, SL: Step Length relative to body height, CM: Center of Mass. *: Significant within-group difference ($p < 0.05$).

(*Alcaraz, Elvira & Palao, 2014*; *Spinks et al., 2007*). However, moderate effect sizes were seen in some early acceleration kinematic parameters, including decreased touchdown CM distance and CM angle in HS50%, corresponding to potentially less time spent in the breaking phase due to contact times not changing. These changes make sense with our cross-sectional sled measurements (Fig. 4), as these were the two variables that showed the largest effect sizes for changes in movement. However, we found no relationships between changes in these variables and improvements in sprint performance, thus more accurate methodological approaches and/or larger sample sizes are likely needed for such short interventions. Furthermore, no negative effects of heavy resisted sprinting were observed on either early acceleration or upright sagittal plane sprint kinematics as speculated to some degree by previous literature (*Alcaraz et al., 2018*; *Alcaraz, Elvira & Palao, 2014*; *Alcaraz et al., 2008*; *Alcaraz et al., 2019*). While both HS60% and CON significantly decreased their contralateral hip angle at touchdown during early acceleration, this was likely due to normal

fluctuations in sprint technique as the result remained under the minimal detectable change (HS60%: 3.13%, CON: 4.01%, MDC: 5.85%), rather than longitudinal alterations caused by the training protocols. One clear explanation is that potential deleterious effects were mitigated via coaching cues targeted to maintain good posture, in place of athletes adopting sub-optimal patterning during the heavy resisted sprinting. Our results cannot support the occurrence of longitudinal technical breakdown following heavy resisted sprint training, or at least indicate that such effects might be reduced with common-sense programming.

As an additional observation, our data showed that initial $F_0$ capacity and sprint FV-profile orientation seems to explain moderately adaptation potential (Fig. 4), corresponding to previous literature (*Lahti et al., 2020*). Thus, if an athlete already has a high force production capacity, or a force-oriented FV-relationship/profile, it should logically reduce adaptation potential to a high force –low velocity stimulus. This sample size does not allow for clear cut-off thresholds for training, however, a recent study using heavy resisted sprints in high-level rugby players showed nearly identical results. Therefore, an initial $F_0$ value around 8.4 N.kg$^{-1}$, or a sprint FV-profile lower than $-0.95$ will likely not respond well to heavy resisted sprint training (*Lahti et al., 2020*). Future studies should explore if varying from individualized (velocity decrement) heavy to light loads based on initial FV-qualities is of further value.

## LIMITATIONS

The control group and the intervention groups were two different teams with inevitable differences in their training culture. Therefore, although initial sprint performance was highly homogenous, differences in training and recovery methods may have contributed to the results. This study also may have been underpowered for some variables, as based on the within- and between-group effect sizes, both groups showed similar trends in early acceleration, but only HS50% reached statistical significance. Furthermore, inclusion of a randomized control group that performs unloaded systematic acceleration training should be compared in future studies. The 2D motion analysis was only based on two time points, therefore caution is advised in their interpretation and future studies are implored to use more rigorous approaches. We did not have access to a high-resolution slow-motion camera, which likely contributed a couple of variables showing lower reliability. Similar to previous resisted sled training literature our sled study used a single time point method (toe-off, touchdown). A more ideal approach would likely be the analysis of waveforms, such as with the statistical parametric mapping method (*Schuermans et al., 2017*). We also acknowledge that the absolute reliability (ICC) confidence intervals can be considered large in numerous analysed variables, making it too imprecise to make accurate conclusions regarding their true reliability. Future studies using similar methods should include a larger sample size to improve reliability measurements.

## CONCLUSION

Providing efficient evidence-based options to enhance sprint performance training is crucial for strength and conditioning coaches in high level soccer settings. It seems that in a time

span of 11 weeks, one of the underlying reasons for heavy resisted sprint training improving sprint performance is increased force production (both directional and absolute). As this took place in a similar step time, the main driver seems to be improved mechanical power and likely rate of force development. Thus, our findings suggest that heavy resisted sprint training can improve sprint performance in professional soccer players. Adaptations may be potentially maximized with a 50% compared to a 60% velocity decrement resistance. A 50% velocity decrement resistance may provide a broader transfer across split-times, which should be verified in future studies. Based on the average amount of resisted sprints that were conducted during this study, the target should be to achieve at least 38 sprints divided over 2 months, preferably 1–2 per week, including a final taper. After familiarization, this stimulus can be integrated efficiently into field conditions, with a session duration lasting ~20 min for the entire team with 4+ sleds. Our results support the assertion that coaches do not have to worry about potential adverse effects on sprint technique if appropriate familiarization, cueing and supervision is used. Furthermore, coaches should be aware that heavy resisted sprint training will very likely not work for the entire team, which can be to some extent predicated by appropriate initial performance tests, including sprint FV-profiling.

## ACKNOWLEDGEMENTS

The authors would like to thank all the athletes and coaching staff that were involved in the study. We would also like to thank the following sports scientists for their consultation to the project and aiding in creating the figures; Andrew Vigotsky, Matt R. Cross, and James Wild.

## ADDITIONAL INFORMATION AND DECLARATION

### Funding
The authors received no funding for this work.

### Competing Interests
The authors declare there are no competing interests.

### Author Contributions
- Johan Lahti and Toni Huuhka conceived and designed the experiments, performed the experiments, analyzed the data, prepared figures and/or tables, authored or reviewed drafts of the paper, and approved the final draft.
- Valentin Romero analyzed the data, authored or reviewed drafts of the paper, and approved the final draft.
- Ian Bezodis and Jean-Benoit Morin conceived and designed the experiments, prepared figures and/or tables, authored or reviewed drafts of the paper, and approved the final draft.
- Keijo Häkkinen conceived and designed the experiments, authored or reviewed drafts of the paper, and approved the final draft.

## Human Ethics

The following information was supplied relating to ethical approvals (i.e., approving body and any reference numbers):

The University of Jyvaskyla Ethical Committee granted ethical approval to carry out the study.

## Data Deposition

The raw data are available as a Supplemental File.

## Supplemental Information

Supplemental information for this article can be found online at http://dx.doi.org/10.7717/peerj.10507#supplemental-information.

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
