# Peer review of "Changes in sprint performance and sagittal plane kinematics after heavy resisted sprint training in professional soccer players"

_PeerJ, doi:10.7717/peerj.10507_

## Round 0.1 · original submission · Major Revisions

Please see the comments from the three reviewers. There was a range of opinions, but on balance I think this paper, with appropriate revisions, will contribute to the body of literature. I look forward to seeing a revised version soon.

Reviewer 1 ·

Basic reporting

No comment

Experimental design

No comment

Validity of the findings

No comment

Additional comments

Changes in sprint performance and sagittal plane kinematics after heavy resisted sprint training in professional soccer players.

The aim of this study was investigated whether heavy resisted training affected sprint performance, kinetics, sagittal plane kinematics, and spatiotemporal parameters in professional soccer players.

The authors performed nine-week protocol of resisted sprint training in professional soccer players with a resistance that induced a 60% velocity decrement from maximal velocity for one group and a second group that used a 50% velocity decrement, and a control group that continuing training as normal with no systematic non-specific acceleration training.

General comments

The positive characteristic of this design is that the intensity was determined taking as reference the decrement of velocity when the subjects trained with a resisted sprint. However, unfortunately, there are many aspects that, in our opinion, invalidate the design.

One of the design problems is that only two levels of independent variable are analysed (60 and 50% velocity decrement). This implies that we cannot deduce which load produces a better positive effect. The lighter load used tends to give a better result, as it seems that this load is the only one that presents some significant differences with respect to the control group. This means that probably lower loads could be better for improving sprint, but this is not analysed in the study, so the design does not provide enough relevant information on resisted sprint training.

Many kinetic and kinematic variables are analysed in the study, but the real effect of interest is the change of velocity in the sprint. Therefore, all other indicators are secondary if the training effect does not translate into more velocity in the sprint.
It seems that the objective of measuring the kinetic and kinematic variables is to "prove" that the loads used do not cause adverse changes in the sprint technique. But if speed is improved, what does it matter whether or not there were adverse changes in technique, especially in team sports players? In addition, the authors state that they found no relationship between changes in some of these variables and improvements in sprint performance (lines 508-09). This commitment to analysing numerous variables in addition to velocity leads the authors to present a disproportionate length of four pages of test results without including tables and figures, while only three pages of discussion.
Very often the authors talk about the differences or not of the kinetic and kinematic variables, when the reliability tests of these measures had very poor results in most cases. In fact, in some cases the value of the lower limit of the 95% CI of the ICC is zero or even negative. We believe that it is not appropriate to make deductions about the effect of variables that are so unreliable.
In relation to the reliability problem, the procedure for calculating the ICC and the SEM is not indicated in the manuscript. This is especially important because the results can be quite different depending on the procedure used. In addition, the ICC values considered as "good" are extremely low, which is related to the values indicated above of zero or negative in the lower limit of the confidence interval.
It is not correct to implement two statistical tests to know the differences between and within groups. You cannot apply an ANCOVA to detect differences between groups and then a "t" test to contrast differences within groups. You should have applied a 2 (measurement) x 3 (group) repeated measurement ANOVA to check differences between and within groups with a single test.
Nor is it appropriate to consider the two experimental groups as a single group for comparison with the control group. It is clear that with only two groups, one experimental with a larger number of subjects, and one control group, it is more likely to find statistically significant differences, but what the authors intend to explain with this analysis. This type of decision does not seem reasonable.
How should we interpret some results as "within-group spatiotemporal analysis showed that HS50% increased maximal velocity step rate (p <0.05, d = 1.50), however, the improvement remained under the minimal detectable change" (Lines 48-50).
The authors decide to apply a sled sprint distance of 0-15 m for the HS60% group and 0-20 m for the HS50% group in order to use a standardized time under tension. This apparent control of an independent variable, which presents equalization problems, introduces the sprint distance as a strange variable, when this variable is easier and more precise to control. Therefore, this does not seem to be an adequate decision if we want to improve the knowledge about the effect of resisted sprint training.

Other comments

Lines 175-77. What was the reason for carrying out all training and testing sessions inside on artificial turf, and made the post testing outside? At best this increases the probability of introducing error into the measures

Lines 192-94 “There was no ordering of the control group, however, the sprint performance was predicted to be similar due to earlier consultation work with the team”

We think this sentence is strange and should have been clarified. What is meant by "team consultation work"?

Lines 225-29 “Load-velocity tests were completed under one unloaded and 3 loaded conditions with one sprint per load (50%, 75%,100% of BM) for both HS groups, outlined in previous literature27. The load-velocity data was then fit with a least-square linear regression to generate an individualized load-velocity profile for each athlete. Thereafter, the individual load corresponding to a 60% and 50%-velocity decrement of maximal velocity was calculated”.

The fact that the tests had been done previously in this way does not guarantee that the procedure is appropriate. If we do not use light loads, the adjustment will always tend to be linear, while including light loads the trend may be different. Therefore, the load corresponding to each decrement of velocity could be different from the one selected in the study

Lines 353-54 “At baseline population variance was not significantly different for any variables”

This statement speaks of pre-training data, but several people were eliminated at the end of the study, so this information should be about the final sample.

Lines 501-03 “The results showed no changes in the kinematics or any other variables in early acceleration, which is in contrast to previous light load literature showing slight increases in trunk lean”

What is the importance of leaning the trunk a little more or not?

Lines 515-24. In our opinion, according to the results of the study and the deficiencies observed in the design, the last paragraph of the discussion seems to us to be highly speculative on the strength or speed profile and the recommendation to train with one load or another

Lines 527-28 and 530-36. The fact that the limitations are exposed is appreciated, but these limitations were already known before the training and are of maximum importance for the design, then the study should not have been done without eliminating most of the limitations.

·

Basic reporting

1. Basic Reporting
The writing is clear and unambiguous, with professional English used throughout. The article includes sufficient introduction and background to demonstrate how the work fits into the broader field of knowledge. Relevant prior literature is appropriately referenced. The structure of the article conforms to an acceptable format of ‘standard sections’. Figures are relevant to the content of the article, of sufficient resolution, and appropriately described and labeled. All appropriate raw data (and more) have been made available.

Further basic reporting comments in approximate order of importance:

1.1. Please proof-read for minor typos. In a number of places letters are missing (for example, showing --> showed in line 45, seem --> seems in line 87, a --> an in line 90, subjects --> subject’s in line 112 and 115…)

1.2. Line 247: There is an inconsistency between for example ‘1.5-m’ here and ’22.5 m’ on line 249. I prefer the space but consistency is the main goal here.

Experimental design

2. Experimental design

This is original primary research within the aims and scope of the journal. The research question is well defined, relevant & meaningful. It is stated how the research fills an identified knowledge gap. The methods are described with sufficient information to be reproducible by another investigator. All data are available as supplementary materials, including all additional reliability analyses and additional information such as training schedules. This facilitates greater inspection of the data and findings by the interested reader and should be commended. The research was conducted in conformity with the prevailing ethical standards in the field.

There are some limitations in the study design (e.g. control group from a different team so not fully matched, small sample size leading to uncertain conclusions based on lack of effect, poor reliability of some measures used, etc.) but these are mostly discussed sensibly and are mostly a consequence of conducting longitudinal training interventions within a professional soccer club.

Further comments in approximate order of importance:

2.1. The statistical power of the study needs to be considered in more detail. A small sample size may be justified on the basis of the population used, but inferences should be made accordingly. This is especially true given that conclusions are drawn on the basis of no significant differences in kinematics. An under-powered study due to small sample size is less likely to identify any true differences and this makes this conclusion more likely. I fully appreciate that sample size is a trade-off when dealing with longitudinal interventions in professional sporting settings, but this should be discussed more explicitly and should be taken into account when drawing conclusions. For example, you could state in the methods what you will consider to be evidence of no difference (evidence of absence rather than absence of evidence). As a guide, please see this recent editorial in Journal of Sports Sciences: https://doi.org/10.1080/02640414.2020.1776002

2.2. There are a lot of tests performed. How did you control for multiple statistical comparisons within the study?

2.3. I really like the use of the minimal detectable change to inform inferences. However, I think the manner in which this is used can be made more objective and be outlined in the methods. For example, are differences only to be considered meaningful if they are greater than the MDC? Does the mean difference or the entire confidence interval need to be greater than the MDC? Clear and consistent standards would improve these inferences. It sometimes seems like the MDC comment is added on to the end of a result statement as an afterthought. As a guide, please see this recent editorial in Sports Biomechanics: https://doi.org/10.1080/14763141.2020.1782555

2.4. Lines 206: Is there any possible benefit to investigating split times such as 10-20 m or 20-30 m to investigate different phases rather than only 0 – x m?

2.5. Line 218: Is a 100% horizontal and 0% vertical GRF direction the optimal?

2.6. Post-testing was performed outside, while pre-testing was performed inside – is there any data to support what difference this would make? Any reliability data for example? If this effect is systematic then this could either blunt the observed pre-post changes or make them seem larger – it is therefore of interest to be quantified.

Validity of the findings

3. Validity of the findings

The data on which the conclusions are based is provided. I was able to quickly recreate some of the figures in the manuscript from the supplied data. As discussed above, the validity (or the reader’s confidence in the validity) could be improved by greater objective detail around the interpretation of: a) magnitudes of difference in relation to the MDC; and b) no significant effect (is this evidence of no effect, evidence of no effect above a certain size (e.g. no meaningful effect), or just an uncertain finding?

Conclusions are well stated, linked to original research question & limited to supporting results.

Additional comments

4. General comments

The aim of this manuscript was to investigate changes in sprint performance and the potential underlying mechanical changes after integrating two different heavy resisted sprint training loading variables within a professional soccer setting. This is a very interesting, novel study with clear practical applications. There are some limitations in the study design (e.g. control group from a different team so not fully matched, small sample size leading to uncertain conclusions based on lack of effect, poor reliability of some measures used, etc.) but these are mostly discussed sensibly and are mostly a consequence of conducting longitudinal training interventions within a professional soccer club. I feel that once any possible adjustments are made the remaining limitations in study design will be an acceptable trade-off for this type of applied sample and intervention. It is extremely unlikely that such an applied question could be answered in a professional sporting setting whilst fully controlling everything – the teams / players would be unlikely to provide access and engage.

Further comments in approximate order of importance:

4.1. Can exact (3 decimal places) p-values be given wherever possible? There are many instances where ‘p < 0.05’ or ‘p > 0.05’ are stated but the exact values would allow better inferences to be made.

4.2. Line 260: What was the reliability of the digitisation process?

4.3. Line 285: Throughout the manuscript, 1-2 sessions per week would be more accurate than 2 per week, especially as the average was around 10-11 in 9 weeks.

4.4. Line 301: I expect this is sprint time rather than ‘time under tension’?

4.5. Line 435: Confidence intervals around the d effect sizes would be useful to indicate the level of precision.

4.6. Line 361: It seems odd to group multiple tests together into one set of confidence intervals – this results in a very wide range being reported.

4.7. Table 2: It may be beneficial to include the MDC within these tables for comparison.

4.8. Line 122: Try to be more specific than ‘unwanted’ – do you mean an increase or decrease in what parameters?

4.9. Line 32: Should ‘heavy resisted training’ include the word ‘sprint’ if it is abbreviated to HS?

Reviewer 3 ·

Basic reporting

All comments are listed below in "General Comments for the authors"

Experimental design

General comment: there are some inherent challenges with conducting a “perfect” experiment when completing a training study (scheduling & logistics, illness & injury, etc.), especially in a professional sports environment. However, it appears the investigators have handled the potential confounding variables in a rigorous manner, and the Methods are comprehensive and provide a good description of the data collection/analysis and intervention protocols.

Validity of the findings

No major issues here. All other comments listed below in "General Comments for the authors"

Additional comments

Overview and Summary Comments:
• This study aimed to evaluate the effects of heavy resisted sprint training in male professional soccer players. This investigation is of high interest because it is rare to have a training intervention with professional athletes as the subject population. Moreover, the improvements in sprint acceleration elicited by the training intervention(s) are notable, both with respect to the improvement in sprint performance with no detrimental effects on kinematics/technique, which has always been a key question regarding heavy resisted sprint training. The authors should be commended for conducting such a comprehensive investigation with intriguing and impactful findings for the fields of sports performance and applied biomechanics.

• Some minor comments and suggested revisions are listed below with the aim of strengthening the manuscript.


Abstract
• Line 33: after “spatial-temporal parameters” it would make sense to add the phrase “in male professional soccer players” to indicate the sex of the subject population, as I don’t think sex of the subjects is mentioned elsewhere in the Abstract.
• Lines 38: the phrase “with no systematic non-specific acceleration training” does not read smoothly to me. Perhaps delete the term ‘non-specific’ (?).

Introduction
• Line 87: minor typo, change “seem” to “seems” or “is”
• Line 106: The second paragraph of the Introduction is somewhat lengthy, perhaps divide into two separate paragraphs to make it easier to follow. I would suggest starting a new paragraph at Line 106: “Interventions with heavy loads…”
• Lines 106-116: perhaps include reference to recent publication by Cahill et al. on resisted sled-pull training in male HS athletes
o Cahill, Micheál et al. (August 2020). Influence of Resisted Sled-Pull Training on the Sprint Force-Velocity Profile of Male High-School Athletes, JSCR. doi: 10.1519/JSC.0000000000003770


Methods
• General comment: there are some inherent challenges with conducting a “perfect” experiment when completing a training study (scheduling & logistics, illness & injury, etc.), especially in a professional sports environment. However, it appears the investigators have handled the potential confounding variables in a rigorous manner, and the Methods are comprehensive and provide a good description of the data collection/analysis and intervention protocols.
• Line 151: To summarize the experimental design at the beginning of this section, I would suggest starting this paragraph with a sentence similar to the following: “A pre-test, post-test experimental design utilizing three training groups was used to examine the effects of heavy resisted sprint training in professional male soccer players.”
• Lines 278-309: If it is possible to provide a little more description on the training activities of the CON group, that may be useful.


Results
• Lines 361-363: is there any explanation or reason suggested for the poor reliability found for the sprint F-V profile slope?
• I would double-check the post-test maximum velocity contact time value for Subject HS60_6 (“Raw Data” excel file, “Sprint Kinematics” worksheet, cell A7.). If I’m interpreting the data correctly, it lists a post-test max velocity contact time of 0.163s for a subject running ~9 m/s, which seems unlikely.
• In Table 3, I would suggest listing the contact time values to three decimal places provide the reader with enhanced resolution/clarity.
• Minor suggestion for Results and Discussion: I would suggest replacing the word “Immediate” with “Acute”

Discussion
• Lines 473-476: This section “However, as Pmax was…” does not read clearly in its current form. Table 2 presents that neither HS60 nor HS50 had statistically significant increases in F0 values compared to CON, but it is also presented that combined HS had increases in F0 that were statistically significantly greater vs. CON (Table 2 and Figure 3A). Based on the latter statistic, it seems that on the whole with both resisted groups combined, HS led to increases in F0 when compared to CON, correct? From this reader’s perspective, the sentences in Lines 473-474 vs. 475-476 appear to contradict each other as presently written. Modified/additional text may help here.
• As listed in Table 3, it’s fascinating that both HS60% and HS50% and significant and large ES improvements in F0 but that HS50 had a slightly better improvement in V0 (albeit not quite significant). In line with this, both HS groups significantly improved Pmax but HS50 to a larger magnitude (ES) than HS60 (although not significantly different HS50 vs. HS60).
o To me, the fact that HS50% had equally large improvements in F0 while still increasing V0 to a larger degree than HS60% lends supporting data to the previous suggestion that HS50% is the optimal load for maximizing acute power but also, evidence that HS50% is the optimal load for longitudinally enhancing power (i.e., per prior work from Cross et al., reference 27). Although it is written that anything in the range of 45-65% velocity decrease could be beneficial (Lines 490-492), the fact that HS50% may indeed be optimal is perhaps something that could be discussed more.

---

## Round 0.2 · Minor Revisions

Thank you for taking the time to respond to the previous comments. After that review there are just a few minor comments that I would like addressed. Please see the reviewer reports for detail. Thank you again.

·

Basic reporting

In relation to a comment and response for Reviewer 1: Reporting the F and df values for the post hoc tests would be a good addition.

Line 254: The brackets around m/body length should be removed.

The plural of subject is subjects. In multiple places subject’s is used incorrectly.

Line 342: The – can be removed between good and excellent as it already says ‘to’

Lines 378-392: Could these p values be reported to 3 decimal places? Especially those that are < 0.1

Experimental design

No further comments on experimental design.

Validity of the findings

Reading through the changes and responses to other reviewers, I believe that the statements (including within the conclusion) around adaptations potentially being maximised with a 50% velocity drop resistance should be downplayed. Your results support statements in some places that 50% velocity drop is more effective than 60% velocity drop but you cannot say that 50% is optimal or likely optimal. The results support that the ‘optimal’ is < 60% but we do not know any more than that from the current study design. The comparison to typical maximal power regions of the force-velocity curve support your claims but it should be made clear where these claims are speculative.

I would still like to see further clarity around the pre-post kinematic changes and the influence of sample size. The lack of kinematic changes is a major finding within the study and so we must ensure that the results support this conclusion. Reporting the actual p values here would be beneficial. Unless I have missed it somewhere (I apologise if this is the case!) it still says simply p > 0.05: ‘No main effects or interaction effects were found for pre and post sprint kinematic variables for both early acceleration and upright sprinting (p > 0.05)’. The reader is therefore unclear whether the p values are 1.00 or 0.051. If the results are not close to significance then this perhaps gives greater confidence that the inferences would not change with a greater sample size. This is important because the lack of significant main effects prevents post hoc tests from being performed. It appears from the confidence intervals (i.e. those that do not cross zero) that had post hoc tests been necessary, some parameters may be significant (or at least very close to significance after controlling for multiple comparison). For example, for 60%: ACC CM angle at touchdown; ACC contralateral hip angle at toe-off and touchdown; MAX CM distance to toe at toe-off and touchdown; MAX CM angle at toe-off and touchdown. I therefore just want to be clear that the results are being interpreted with the necessary caution. I believe greater reporting of exact p values / effect sizes will help with this, along with discussion of the level of (un)certainty around these findings. Whether there is or is not a change in kinematics, this should not affect the ability to publish. It is simply important to make sure the findings are clear. To be clear, I am not suggesting that you perform post-hoc tests if the main effects do not support these - I just wish to see a bit more information to help evaluate the likelihood of a type II error and would like to see this discussed when results are interpreted and conclusions formed. It is likely that the effect sizes in kinematics may differ to those in kinetics or split times. These likely effect sizes (from previous studies) could also support the sample size and / or the interpretation of non-significant results.

I still believe that the conclusion should reflect the 1-2 sessions per week performed in the study. I don’t believe you can say ‘preferably twice per week’ when your study does not show this to be preferable compared to once per week. 1-2 or ‘at least once’ reflects the study more accurately in my opinion. It is important that the conclusions and practical recommendations are supported by the study – a lot of the conclusion paragraph is detailing what was done in the study, rather than parameters that were systematically perturbed and investigated. You can therefore say that this type of intervention is successful in improving certain parameters. However, I think you need to be clear where your recommendations are based on a single effective method and we do not know if they are the most effective method (e.g. number of sessions per week, session duration, number of sprints, taper period, etc.).

Additional comments

Thank you for your detailed response and the many changes made to the manuscript. I believe the manuscript has improved and I hope that my few remaining comments can be relatively easily addressed.

Your statements / general principle of a more horizontal GRF being better is valid as long as a greater % horizontal is always beneficial within the range of values reported. From the values in the table this is likely the case. I suggest including a statement such as ‘A more horizontally oriented ground reaction force was considered beneficial within the range of values reported in this study’.

Reviewer 3 ·

Basic reporting

The writing and basic reporting is clear and unambiguous, with appropriate references and sufficient background and context provided. Overall the updated version of this manuscript reads very smoothly. The authors have made the edits that I suggested as it relates to the writing/text in the manuscript.

Experimental design

The research question is well defined, and quite important from a practical standpoint in the fields of applied biomechanics and sports performance. The manuscript clearly defines how this research fills a gap in the current literature. Although challenges exist in conducting training/intervention studies with professional sports teams, the authors have identified these limitations, and from my perspective these limitations do not outweigh the contributions of this study or detract from the impact of the findings.

Validity of the findings

In the revised version of the manuscript, the authors have revisited the statistical approach based on comments of the other reviewers. From my perspective, the current statistical analysis and reporting is robust and allows for straightforward understanding of the results. The conclusions presented in the Discussion nicely follow the results, with an appropriate amount of interpretation and speculation.

Additional comments

In this updated version of the manuscript, the authors have made the necessary modifications to my comments and suggested revisions. Overall, the revised manuscript is stronger than the originally submitted version, with important practical applications for coaches and athletes.

---

## Round 0.3 · accepted · Accept

Thank you for your thorough review of this paper. I appreciate the extra effort you took to clarify the statistical findings too.